# Genome-Wide Identification of the Lectin Receptor-like Kinase Gene Family in *Avena sativa* and Its Role in Salt Stress Tolerance

**DOI:** 10.3390/ijms252312754

**Published:** 2024-11-27

**Authors:** Gui Xiong, Dongli Cui, Yaqi Tian, Trude Schwarzacher, John Seymour Heslop-Harrison, Qing Liu

**Affiliations:** 1Laboratory of Plant Resources Conservation and Sustainable Utilization, South China Botanical Garden, Chinese Academy of Sciences, Guangdong Provincial Key Laboratory of Applied Botany, Guangzhou 510650, China; xionggui23@scbg.ac.cn (G.X.); donglicui2018@163.com (D.C.); tianyaqi24@mails.ucas.ac.cn (Y.T.); 2Key Laboratory of National Forestry and Grassland Administration, Plant Conservation and Utilization in Southern China, Guangzhou 510650, China; 3University of Chinese Academy of Sciences, Beijing 101408, China; 4South China National Botanical Garden, Guangzhou 510650, China; ts32@le.ac.uk; 5Department of Genetics and Genome Biology, Institute for Environmental Futures, University of Leicester, Leicester LE1 7RH, UK

**Keywords:** *Avena sativa*, salt stress tolerance, transcriptome, lectin receptor-like kinases, expression pattern

## Abstract

Lectin receptor-like kinases (LecRLKs) are membrane-bound receptor genes found in many plant species. They are involved in perceiving stresses and responding to the environment. Oat (*Avena sativa*; 2*n* = 6*x* = 42) are an important food and forage crop with potential in drought, saline, or alkaline soils. Here, we present a comprehensive genome-wide analysis of the *LecRLK* gene family in *A. sativa* and the crop’s wild relatives *A. insularis* (4*x*) and *A. longiglumis* (2*x*), unveiling a rich diversity with a total of 390 *LecRLK* genes identified, comprising 219 G-types, 168 L-types, and 3 C-types in oats. Genes were unevenly distributed across the oat chromosomes. GFP constructs show that family members were predominantly located in the plasma membrane. Expression under salt stress demonstrated functional redundancy and differential expression of *LecRLK* gene family members in oats: 173 members of this family were involved in the response to salt stress, and the expression levels of three C-type genes in the root and leaf were significantly increased under salt stress. The results show the diversity, evolutionary dynamics, and functional implications of the *LecRLK* gene family in *A. sativa*, setting a foundation for defining its roles in plant development and stress resilience, and suggesting its potential agricultural application for crop improvement.

## 1. Introduction

During evolution, plants develop a series of mechanisms to maintain oxidative balance under various stressors [1]. All kinds of receptors located on the cell membrane or in plant cells can specifically recognize bioactive molecules and receive and process external information [2]. The receptor-like kinases (RLKs) are an important and diverse group of cell surface receptor proteins that possess an extracellular domain, a membrane-spanning domain, and an intracellular kinase domain [3]. Based on the variability of extracellular structural domains, *RLKs* can be classified into 17 subgroups [2,4]. Lectin receptor-like kinases belong to the *RLK* gene family, which was named for its lectin/lectin-like ectodomain which can bind carbohydrates [5]. The N-terminal lectin structural domain allows LecRLKs to recognize environmental stimuli, whereas the C-terminal intracellular kinase structural domain phosphorylates downstream proteins to transmit signals. Depending on the type of lectin domain at the N-terminus, LecRLKs are further classified into three subfamilies: L-type, G-type, and C-type LecRLKs [3]. L-type LecRLKs possess a legume-like lectin domain. G-type LecRLKs have an α-mannose-binding bulb lectin domain, along with an S-locus glycoprotein domain (SLG) and plasminogen apple nematode (PAN) and/or epidermal growth factor (EGF) domains. The SLG domain has been proved to play an important role in the self-incompatibility of plant gametophytes [6]. The PAN motif is associated with protein–protein and protein–carbohydrate interactions, while the EGF domain likely contributes to the formation of disulfide bonds [7]. C-type LecRLKs are marked by a calcium-dependent lectin domain [3,5]. LecRLKs have been identified and discovered in various plants, including 75 members in *Arabidopsis thaliana*, 173 members in rice [4], and 263 members in wheat [8].

LecRLKs play a crucial role in plant growth, stress management, and innate immune responses [9,10,11]. Crop yield loss due to soil salinization is an escalating threat to global agriculture [12]. Salt stress, either due to soil type or from irrigation water, impacts the growth, development, and grain productivity of crops [13] in arid and semi-arid regions [14]. With increased irrigation, global warming, and rising sea levels, the area of land (including reclaimed land) under salt stress is increasing, while the use of saline land for productive agriculture has become a new trend in the development of agriculture and animal husbandry [13,15].

Many plant processes involve families of genes with identical or related functions, or within a pathway. For example, the glucanase gene family plays a defensive role in response to pathogen attacks [16]; the flavonoid gene family plays a key role in plant growth and development, response to adversity, and interactions with the environment [17]. In some pathways, gene clusters play a role in the specialized metabolism of plants [18]; for the flavonoid pathway, the presence of gene clusters contributes to the concentration of members of different gene families in physical locations [19,20,21]. The study of gene families, genes in a pathway, and gene clusters not only improves the understanding of plant metabolic pathways and their dynamics, regulation, and evolution, but also provides important molecular resources for crop improvement and the breeding of new varieties.

Plants have developed regulatory mechanisms to cope with salt stress over a long period of time in response to the environment, including regulation of salt stress resistance through maintenance of photosynthesis, ion homeostasis, and synthesis of sugar metabolism, proline, and other compatible solutes that regulate reactive oxygen species (ROS) levels [13,22]. Several reports have presented the involvement of LecRLKs in plant responses against salt stress. A G-type LecRLK known as GsSRK in *Medicago sativa* has the potential to regulate Na^+^/K^+^ balance during salt stress, possibly through the scavenging of reactive oxygen species (ROS) and the modulation of osmotic homeostasis [23,24]. Under salt stress conditions, *SIT1* (an LecRLK gene) enhances the sensitivity of plants [25]. The overexpression of one *Pisum sativum* LecRLK, PsLecRLK, prevents the accumulation of reactive oxygen species and membrane damage and enhances salt tolerance in plants [9].

Here, we aimed to perform a comprehensive genome-wide identification and characterization of the lectin receptor-like kinase (LecRLK) gene family in *Avena sativa* (oat), with a focus on elucidating their potential roles in salt stress tolerance. Specifically, this study aimed to identify *LecRLK* gene family members and analyze their gene structures, conserved domains, and phylogenetic relationships. Additionally, utilizing transcriptomic data, we investigated the expression profiles of these genes under salt stress conditions. Key candidates were further validated through RT-qPCR analysis and subcellular localization assays, enhancing our understanding of their functional roles in mediating salt stress responses. This work provides new insights into the molecular mechanisms underlying salt stress tolerance in oats and offers valuable resources for future breeding efforts aimed at improving salt tolerance in this crop.

## 2. Results

### 2.1. Identification of LecRLK Gene Family Members in Oat

A total of 390 putative AsaLecRLK protein sequences were identified in *A. sativa*, which were further classified into L-type LecRLKs (n = 168), G-type LecRLKs (n = 219), and C-type LecRLKs (n = 3). The physicochemical properties of AsaLecRLK family genes (named L, G, and C then in order on chromosomes) were summarized (Appendix A).

Coding sequence size (CDS) length ranged from 915 to 2943 bp, 1338 to 2976 bp, and 1662 to 1680 bp, whereas the protein lengths varied from 304 to 980 aa, 445 to 991 aa, and 553 to 559 aa for L-type, G-type, and C-type proteins, respectively. The molecular weights ranged from 34.10 kDa (AsaLecRLK-L-type-168) to 108.43 kDa (AsaLecRLK-G-type-200), and the isoelectric point (pI) ranged from 4.53 (AsaLecRLK-G-type-28) to 10.20 (AsaLecRLK-L-type-168). In total, 281 LecRLK proteins were acidic (pI < 7); the remaining 106 proteins were alkali (pI > 7). The grand average of hydropathicity (GRAVY) ranged from −0.311 to 0.102, with 344 proteins exhibiting hydrophilicity (GRAVY < 0) and 46 proteins exhibiting hydrophobicity (GRAVY > 0), indicating diverse functions among this protein family. Subcellular localization prediction results were analyzed using the program WoLF PSORT, which showed that 222 proteins were only located in the plasma membrane, and the other 168 were in different cellular compartments, including extracellular, plastid, vacuole, chloroplast, mitochondria, cytoplasm, nucleus, etc. (Appendix A), indicating the diverse roles of the LecRLK family in variable environments.

### 2.2. Chromosomal Distribution and Phylogenetic Analysis of LecRLK Gene Family Members

The chromosomal positions of AsaLecRLKs showed that 390 members of the gene family were distributed (Figure 1) across all 21 chromosomes and the unanchored scaffold. *AsaLecRLK* genes are less abundant in regions with low gene density. The numbers of genes per chromosome ranged from 39 (chromosome 2D) to 9 genes (1C). Gene clusters were observed on each chromosome, perhaps a consequence of segmental duplications.

Evolutionary relationships of the AsaLecRLK members were shown in a phylogenetic tree, with outgroups of three animal Pelles [26] [the animal homologs of Arabidopsis RLKs; a Pelle kinase (DmPelle) in Drosophila, a Pelle-like kinase (CePelle) in Caenorhabditis, and an IRAK in humans], and 390 LecRLKs of *A. sativa* (maximum-likelihood (ML) method; Figure 2). The result showed that the L-type, G-type, and C-type formed monophyletic lineages.

### 2.3. Conserved Domain, Conserved Motif and Gene Structure Analysis of AsaLecRLKs

Conserved domain analysis of AsaLecRLKs demonstrated that there were significant differences in the structures of *AsaLecRLK* genes within and between the three subgroups (Figure 3 and Appendix A). There were six conserved domains in the G-type AsaLecRLKs, including the protein kinase domain (Pkinase; PF00069), S-locus glycoprotein domain (S_locus_glycop; PF00954), mannose-binding bulb-lectin domain (B_lectin; PF01453), two PAN domains, and a DUF3403 domain. Based on the type and number of conserved domains, the G-type AsaLecRLKs can be classified into seven distinct categories: a total of 33 G-type AsaLecRLKs exhibited two domains including Pkinase and B-lectin, and 20 contained Pkinase with B-lectin and S_locus_glycop. Additionally, 145 genes contain an additional PAN domain, either PAN_1 or PAN_2, beyond the three aforementioned structural domains. A total of four G-type AsaLecRLKs contain the DUF3403 domain at their C-terminus. Fifteen genes are identified to contain a lectin domain, a kinase domain, and a PAN domain. One gene featured two kinase domains. Within these domains, the S-locus_glycoprotein is implicated in the self-incompatibility response in plants; the PAN domain is involved in protein–protein and protein–carbohydrate interactions; and the DUF3403 domain has been previously reported in the literature, although its function remains elusive. In addition, transmembrane domain predictions showed that the majority of AsaLecRLK proteins (96.92%) had transmembrane domains, which is consistent with previous reports in other species. The L-type and C-type have a less complex domain structure: 166 L-type AsaLecRLKs contain a single legume lectin domain (Lectin_legB; PF00139) and Pkinase. Some unique domain architectures were also observed within the L-type AsaLecRLKs: two L-type AsaLecRLKs contain an adh_short domain (PF00106), which is involved in a variety of oxidation–reduction reactions. There was only one calcium-binding lectin (Lectin_C; PF00059) and one Pkinase domain present in three C-type AsaLecRLKs.

Ten conserved motifs in the *AsaLecRLK* gene family were identified using MEME online tools [27] and ten motifs were detected in 390 AsaLecRLK proteins from *A. sativa* (Appendix A). The results showed that most of these existed in the kinase domain. This confirms that the kinase domain has remained conserved during evolution. All G-type genes of the AsaLecRLK family had B-lectin with the typical motif 10, which is absent in the L-type and C-type genes. AsaLecRLK proteins of the same type had similar domains to the conserved motifs (Appendix A) and distributions, indicating functional similarity among members of the same type. The variation in motifs among different types also demonstrates the functional diversity of LecRLKs.

The gene structure of the *AsaLecRLK* gene family showed structural diversity. The number of *LecRLK* gene exons of *A. sativa* ranged from 1 to 9 (0 to 8 introns), although most (215 out of 390) genes had a single exon. Intron length varied from 61 bp (*AsaLecRLK-G-type-106*) to 2041 bp (*AsaLecRLK-L-type-27*) (Appendix A), with subfamily branches showing similarities in structure.

### 2.4. Cis-Acting Element Prediction of AsaLecRLKs

Promoter cis-element analysis provided insights into the tissue-specific expression and stress response patterns of the gene family. Except for the TATA-BOX and CAAT-BOX, a total of 120 cis-acting elements were identified, with a variety of environmental and stress response elements in the *AsaLecRLK* gene family of *A. sativa* (Appendix A), using the PlantCARE [28] analysis of the upstream 1500 bp sequence. The analysis associated these elements with wound response, hormone response, light response, promoter and enhancer elements, binding-site elements, and development. Among them, the most abundant element is transcription factor MYB (1086), which has been identified in plant development and responses to stress by combining with MYB cis-elements in promoters of target genes [29]. The second most abundant element is MYC (1054), which is associated with dehydration response and ABA induction [30]. The third most abundant element is ABRE (978), which is associated with hormone response [31]. The fourth most abundant element is STRE (837), which is associated with environmental stress response [32,33]. Additionally, the elements as-1, CGTCA-motif, and TGACG-motif were abundant, all of which are hormone-responsive elements, implying that the expression of *AsaLecRLKs* might be regulated by phytohormones [30,32]. Among the *AsaLecRLK* gene family, AsaLecRLK-L-type-162 has the highest number of cis-acting elements (86).

### 2.5. Expansion and Selection Pressure Analysis of AsaLecRLKs

Gene family formation, member expansion, and functional diversification are primarily driven by the process of gene duplication. Among 390 *AsaLecRLK* genes, we found that there were 190 possible pairs of segmental duplication (SD) genes and 19 pairs of potential tandem duplication (TD) genes (Appendix A). Among these 19 tandem duplication gene pairs, we identified the formation of a gene cluster of tandem duplication on the Asa7D and Asa4D chromosomes. The results of gene duplication analysis indicated that SD and TD were the main expansion mechanisms of the *AsaLecRLK* gene family. In addition, we calculated the non-synonymous (Ka) and synonymous (Ks) substitution rates, as well as the Ka/Ks ratios. The Ka/Ks ratio varied from 0 to 0.862761 (Appendix A), indicating that purification selection plays an important role during geneexpansion.

### 2.6. Collinearity Analysis of AsaLecRLK Genes

To explore the evolutionary mechanism between the *AsaLecRLK* genes of *Avena sativa* and other species, synteny analyses were performed on four representative plants, including two possible ancestral species of oats (*A. longiglumis* and *A. insularis*), a monocotyledon (*Oryza sativa*), and a dicotyledon (*Arabidopsis thaliana*). Dual synteny analysis revealed that 246 and 356 *AsaLecRLK* syntenic gene pairs were identified in *A. longiglumis* (Appendix A) and *A. insularis* (Appendix A), respectively, and 116 and 24 syntenic gene pairs each were identified in *Oryza sativa* (Appendix A) and *Arabidopsis thaliana* (Appendix A). Consistent with known relationships, these results indicated that oat was most distantly related to *Arabidopsis* and most closely related to *Avena insularis*. By comparing the collinear relationship between*A. sativa*, *A. insularis*, and *A. longiglumis*, it was found that collinearity did not only occur on homologous, but also heterologous chromosomes (Figure 4), supporting that oat chromosomes have undergone chromosomal rearrangement and gene duplication events during evolution.

### 2.7. Gene Expression Analysis of AsaLecRLKs’ Response to Salt Stress Treatments

To investigate the role of AsaLecRLKs in oat salt stress resistance, we analyzed RNA-sequencing data, focusing on the RPKM-based expression patterns of *AsaLecRLK* genes in the root and leaf to provide insights into their potential functions. We analyzed the expression of the 390 *AsaLecRLK* genes under salt stress, first removing 217 genes with FPKM less than one in both root and leaf before and after salt stress. We generated a heat map of the 173 (G-type: 96; L-type: 74; C-type: 3) expressed *AsaLecRLK* genes using the FPKM values (Figure 5). Among these genes, three of the C-type *AsaLecRLKs* were up-regulated in both the root and leaf under salt stress. In total, 45 G-type genes were down-regulated in the leaf and 50 G-type genes were up-regulated in the leaf; 58 G-types were down-regulated and 37 up-regulated in the root. A total of 34 G-type genes were down-regulated in both the root and leaf, and 27 G-type genes were up-regulated in both the root and leaf. A total of 25 L-type genes were down-regulated in the leaf and 47 L-type genes were up-regulated in the leaf; 30 L-type genes were down-regulated and 42 L-type genes were up-regulated; and 13 L-type genes were down-regulated in both the root and leaf, and 62 L-type genes were up-regulated in both the root and leaf. The results indicate the differential expression of some *AsaLecRLK* genes in the root and leaf, with genes such as *AsaLecRLK-G-type-45/68*, *AsaLecRLK-L-type-70*, and *AsaLecRLK-C-type-2* being expressed more in the root than in the leaf; different *AsaLecRLK* genes exhibit varying expression levels under identical salt stress conditions, and their expression also differs between the root and leaf. A cluster analysis showed that *AsaLecRLK* genes located in the same evolutionary branch showed some similarity. For example, *AsaLecRLK-G-type-45* and *AsaLecRLK-G-type-68* showed similar expression patterns after NaCl treatment. However, there is also disaggregation of genes located in the same evolutionary branch. The differential expression profiles among homologous genes suggest that during the evolution of oats, some genes may have acquired new functions or lost old ones following polyploidization. This phenomenon is referred to as the subfunctionalization of homologous genes by some researchers [34].

To verify the results acquired from RNA-Seq data, we selected 19 *AsaLecRLK* genes from three different subtypes for RT-qPCR analysis. The results showed that the expression patterns of these genes were basically consistent with the RNA-sequencing results after 48 h of salt stress, indicating that these 19 genes responded to salt stress to different degrees. Under salt stress treatment, there were significant differences in the expression of multiple *AsaLecRLK* genes in the root and leaf of *Avena sativa* at different time points (0 h, 6 h, 12 h, 24 h and 48 h). Overall, most genes showed different degrees of up-regulation or down-regulation in both root and leaf tissues, with some genes showing significantly enhanced or attenuated expression at specific time points (such as the significant increase in expression of *AsaLecRLK-C-type-3* in the root at 24 h). Additionally, there were tissue-specific differences in the response of root and leaf tissue to salt stress at different time points, suggesting that these genes may have potential tissue-specific functions in the response to salt stress. For instance, the expression levels of *AsaLecRLK-L-type-7/8* and *AsaLecRLK-G-type-200* genes decreased in root tissue but increased in leaf tissue under stress at the 6 h time point, which may be closely related to their functions and participation in *A. sativa* salt stress at different stages. Furthermore, the timing of peak expression levels differs between root and leaf tissue. For example, the *AsaLecRLK-G-type-110* gene reaches its peak expression in the root 48 h post-stress, whereas in the leaf, the peak is attained 24 h post-stress (Figure 6). The presence of temporal differences in the peak appearance of these genes after salt stress suggests that these genes may play different roles at different times, and it is hypothesized that they interact in response to salt stress. Notably, the expression of all three C-type *AsaLecRLKs* was significantly increased in root tissue after salt stress, suggesting that all three genes may be involved in salt stress.

RNA-sequencing and RT-qPCR analyses revealed the spatiotemporal expression patterns of *AsaLecRLK* genes in oat under salt stress, demonstrating significant tissue specificity and temporal dependency. The differential peak expression levels observed in root and leaf suggest that *AsaLecRLK* genes may contribute to salt stress tolerance through tissue-specific and time-dependent mechanisms, potentially involving synergistic interactions. These findings underscore the potential functional importance of *AsaLecRLK* genes in regulating salt stress responses in oat.

### 2.8. Subcellular Localization Analysis of AsaLecRLK-G-type-45

In this study, most of the AsaLecRLK proteins were predicted to be localized to the plasma membrane. To validate the subcellular localization predictions and clarify the functional sites of AsaLecRLK proteins, we cloned the coding sequences of AsaLecRLK-L-type-43 (2043 bp long), AsaLecRLK-L-type-44 (2043 bp), and AsaLecRLK-G-type-45 (2550 bp). The subcellular localization result showed that AsaLecRLK-L-type-43, AsaLecRLK-L-type-44, and AsaLecRLK-G-type-45 were localized to the plasma membrane, which was consistent with the predicted results (Figure 7). The 35S::GFP, used as a control, was localized to both nuclear and cell membranes.

## 3. Discussion

As receptor-like kinases at the cell surface, LecRLKs have been demonstrated to play a crucial role in enhancing plant resistance against abiotic (salt, low-temperature, or drought stress and mechanical damage) and biotic (plant disease caused by bacteria, viruses, fungi, and herbivorous insects) stresses as well as in development (seed germination, leaf development, flower organ development, and reproductive development) [35,36]. Here, we identified 390 genes encoding LecRLK proteins in oat, *A. sativa* (Figure 1 and Figure 2, Appendix A). The number of *LecRLK* genes in the hexaploid oat was higher than in diploid reference species (e.g., 173 in *Oryza sativa* [4]; 113 in *Hordeum vulgare* [31]; 75 in *Arabidopsis thaliana* [4]), with clear duplication between homoeologous chromosomes (Appendix A center); and between rice and Arabidopsis (Figure 4A,B).

Gene tandem replication and fragment replication events are considered to be the key mechanisms for increasing gene family diversity. Analysis based on duplication events indicated that segmental duplication was the major factor leading to the amplification of the *AsaLecRLK* gene family. At the same time, 19 tandem duplication pairs were identified in oat, and interestingly, these tandem duplication gene pairs were not strictly clustered in the phylogenetic tree, suggesting that these *AsaLecRLK* genes may have evolved different functions after duplication [37] (Figure 2, Appendix A). *Avena sativa* is an allohexaploid, and the diploid species *A. longiglumis* and the tetraploid species *A. insularis* are its potential ancestral relatives [38]. We identified 219 G-type LecRLK members in oat, which exceeds the combined total of 205 in *A. longiglumis* (Figure 4C) and *A. insularis* (Figure 4D). In contrast, the 168 L-type members in oats are fewer than the combined total of 176 in the two ancestral species. The number of C-type members is equal to the combined total, with three in each. This result suggests that *LecRLK* genes in oat may have undergone gene loss (or degeneration) and duplication during the evolutionary hybridization and polyploidization events. This study found that *AsaLecRLK* family members are unevenly distributed across the A (185), C (116), and D (142) subgenomes (Figure 1), not least as a consequence of chromosomal rearrangements between genomes in the hexaploidy [39].

By analyzing the predictions of domain architecture and organization, we observed several interesting features of AsaLecRLKs. Firstly, the majority of AsaLecRLKs possess transmembrane (TM) domains. Additionally, we have identified several lectins with two or three transmembrane domains. It is speculated that AsaLecRLKs function as potential membrane-bound receptors (Figure 7) and may bind to the membrane system in a variety of ways [37]. Interestingly, an adh_short domain was identified at the C-terminus of AsaLecRLK-L-type-66 and AsaLecRLK-L-type-27. The adh_short domain belongs to short-chain dehydrogenase/reductase gene family (Figure 3 and Appendix A), which is one of the largest and oldest NAD(P)(H)-dependent oxidoreductase families [40]. It has been demonstrated that LecRLKs can recognize DNA; LecRK-VI.2 is a potential receptor for extracellular NAD+ (eNAD+) and NAD+ phosphate (eNADP+) [41], suggesting that AsaLecRLK-L-type-66 and AsaLecRLK-L-type-27 may also be potential receptors for NAD+.

The gene structure (Appendix A) and conserved motifs (Appendix A) of AsaLecRLKs were analyzed. All G-type AsaLecRLKs have motif 10, indicating that motif 10 plays a crucial regulatory role in G-type AsaLecRLKs. The conserved motif 10 is unique in all G-type RLKs, and is specifically located within the B-lectin domain. This suggests its potential involvement in key functional processes, possibly related to signal transduction or stress response pathways specific to this subfamily. This conserved motif might be a target for further functional analysis to elucidate its role in salt stress tolerance. Intron gain or loss and intron density significantly impact the evolution of large eukaryotic genomes [42]. Gene structural analysis reveals that more than half of *AsaLecRLKs* lack introns. Genes with similar numbers of exons and introns that cluster together on a phylogenetic tree may indicate shared ancestry or conserved gene structures that have been maintained throughout evolution [43]. This conservation could suggest that these genes are subject to similar functional constraints or are involved in similar biological processes. The clustering can also reflect the evolutionary history and potential subfunctionalization or neofunctionalization events that have occurred after gene duplications [44].

Soil salinization represents a significant stressor that adversely affects the majority of plants. Extensive research has established that plants possess the ability to perceive salt stress signals and promptly activate signaling pathways, thereby re-establishing cellular homeostasis through the regulation of growth and metabolic processes [45]. For example, encoding transcriptional factor OsWRKY53 and mitogen-activated protein kinase OsMKK10.2 mediate root Na^+^ flux and Na^+^ homeostasis [46]. In addition to regulating ion homeostasis, salt stress induces enzymatic and non-enzymatic systems in plants to mitigate reactive oxygen species (ROS) stress [47]. Enzymatic scavengers include superoxide dismutase, ascorbate peroxidase, catalase, guaiacolperoxidase, guaiacol peroxidase, dehydroascorbate reductase, monodehydroascorbate reductase, glutathione peroxidase, and glutathione S-transferase. Salt stress also triggers abscisic acid (ABA) pathways. Salt treatment increases ABA concentrations in plant cells and activates sucrose non-fermenting 1-related protein kinase 2 (SnRK2) kinase activity [15,48].

LecRLKs play a role in tolerance to salt stresses [2]. A rice L-type LecRLK gene, *SIT1*, is activated during salt stress, phosphorylates downstream effectors, triggers ethylene signaling-induced reactive oxygen species accumulation, and enhances plant sensitivity [25]. The overexpression of the rice *OsSIK2* gene leads to enhanced salt stress tolerance in plants and promotes leaf emergence [49]. In 2020, Passricha et al. [50] demonstrated that *OsLecRLKs* play a crucial role in salt stress responses. Overexpression of *OsLecRLKs* activated sucrose non-fermenting-related kinase-1 (SnRK1), which in turn activated the salt overly sensitive 1 (SOS1) channel to transport Na^+^ ions out of the cytosol, maintaining ionic balance. In contrast, down-regulation of *OsLecRLKs* led to cytosolic Na^+^ accumulation, disrupting ionic homeostasis and normal cell function. In soybean, a G-type lectin receptor-like kinase GsSRLK positively regulates plant tolerance to salt stress [24]. Due to the close homologous relationship between *SIT1* and *AsaLecRLK-L-type-26*, *AsaLecRLK-L-type-43*, *AsaLecRLK-L-type-67*, *AsaLecRLK-L-type-118*, *AsaLecRLK-L-type-128*, and *AsaLecRLK-L-type-138*, it is speculated that these *AsaLecRLKs* may be related to salt stress. In our study, all selected RT-qPCR genes exhibited varying degrees of response to salt stress, with the *AsaLecRLK-G-type-110* gene demonstrating a 15-fold increase in expression in root tissue 48 h post-salt stress and an 18-fold increase in leaf tissue 24 h post-salt stress (Figure 6). The cis-acting elements of the gene promoter regions are involved in the regulation of gene expression. In this study, the promoter sequences of the *AsaLecRLK* genes predicted multiple cis-acting elements related to stress responsiveness (STRE, TC-rich repeats), phytohormone responsiveness (TGAGG-motif, CGTCA-motif, as-1, ABER, TATC-box, TCA-element), drought-inducible MYB binding site elements (MBS, MYB, Myb, MYC, Myc), and plant growth and development (CAT-box) (Appendix A). Among the identified cis-acting elements, those involved in hormone response are predominant, suggesting that there is a close regulatory link between hormone signaling pathways and the activation of AsaLecRLKs in response to environmental stress. LecRLKs have been shown to play an important role in plant responses to biotic and abiotic stresses and innate immunity. The signaling pathways of the *LecRLK* gene family and their potential ligands have lacked systematic study, and so far, only four ligands of LecRLKs have been identified (eATP, eNAD+, eNADP+ and 3-OH-C10:0) [51], and very little is known about the other ligands, so more studies are needed in the future to explore the regulatory mechanisms. The use of single-molecule analyses to analyze the amino acid sequence of proteins (e.g., Nanopore) offers a novel perspective for studying protein structure and function [52] in combination with the improved computational approaches to analysis of protein structures and interactions with ligands (e.g., Deepmind and AlphaFold3 [53]). Detection of kinase activity and post-translational modification sites defines structural and functional diversity in proteins and identifies downstream substrates. With functional information related to sequences, gene editing technologies such as CRISPR-Cas9 can be employed to regulate the expression of LecRLKs, enabling the development of stress-resistant, high-efficiency crop varieties. For example, Wang et al. [54]. demonstrated that knocking out the L-type LecRLK gene *OsCORK1* enhances rice tolerance to copper stress. Polysaccharide microarrays can be used for high-throughput screening of protein–polysaccharide interactions, which is crucial for understanding processes such as cell signaling, immune responses, and pathogen infection. Techniques like yeast two-hybrid and co-immunoprecipitation can be utilized to study the interaction networks of LecRLK proteins, shedding light on their mechanisms of action in cellular signal transduction.

With global environmental change, biotic and abiotic stresses are becoming increasingly critical for plant growth and crop yield, and the regulatory mechanisms mediated by LecRLKs have the potential to enable plants to meet these challenges. For oats, a high-quality human food and animal feed, there are huge opportunities to improve the crop and overcome the significant constraint on oat cultivation caused by progressive increase in soil salinity due to irrigation with brackish water. The utilization of key genes from the *Avena* pangenome provides a promising avenue for improving the oat crop’s tolerance to salinity stress, a critical advance for achieving sustainable agricultural practices.

## 4. Materials and Methods

### 4.1. Database Search and Retrieval of Lectin Receptor-like Kinase (LecRLK) Protein Sequences in Avena sativa, Avena insularis, and Avena longiglumis

The genomic resources of the hexaploid oat *Avena sativa* cv. ‘Sang’ and the diploid *A. longiglumis* accession PI 657387 [55] were acquired from the National Center for Biotechnology Information (NCBI) database (https://identifiers.org/ncbi/insdc.gca:GCA_030063025.1, accessed on 11 September 2023). The genomic resources of the tetraploid *A. insularis* cv ‘BYU209’ were acquired from the GrainGenes database (https://wheat.pw.usda.gov/GG3/content/avena-insularis-download, accessed on 11 September 2023) [56].

The Hidden Markov Model (HMM) profiles of the kinase domain (PF00069), N-terminal domain B_lectin (PF01453), Lectin_legB (PF00139), and Lectin_C (PF00059) were downloaded from the InterPro database (https://www.ebi.ac.uk/interpro/entry/pfam/, accessed on 21 January 2024) [57]. We retrieved genes containing a kinase domain by running the program hmmsearch (HMMER v.3.3.2) [58] with default parameters to search the kinase profile (PF00069) within three genome protein sequences. Then, all the candidate protein sequences were submitted to the National Center for Biotechnology Information by the batch CD-search tool (https://www.ncbi.nlm.nih.gov/Structure/bwrpsb/bwrpsb.cgi, accessed on 23 January 2024) [59] and online software InterPro (https://www.ebi.ac.uk/interpro/entry/pfam/, accessed on 27 January 2024) to verify the conserved domains.

The protein was predicted by TMHMM Server v. 2.0 (http://www.cbs.dtu.dk/services/TMHMM/, accessed on 2 February 2024) [60] based on a deep learning model to determine whether it was a membrane protein. Subcellular localization was predicted using WoLF PSORT (https://wolfpsort.hgc.jp/, accessed on 4 February 2024). The molecular weight, theoretical isoelectric point, and grand average of hydropathicity of each LecRLK protein were obtained using the ExPasy website (http://au.expasy.org, accessed on 7 February 2024). The amino acid length and coding sequence length (bp) of each LecRLK protein were obtained using the software SeqKit v.2.3.0 [61].

### 4.2. Phylogenetic and Alignment Analysis

Using MEGA-11 (https://www.megasoftware.net/; accessed on 15 February 2024), multiple amino acid sequence alignment (MSA) was performed by the ClustalW [62] algorithm with default settings. Phylogenetic analysis of the LecRLK gene family was constructed with maxlikehood methods by using the FastTree v.2.1 with default settings [63]. The resulting phylogenetic tree was visualized by using iTOL v.4 [64]. Basing on the classification method used for *A. thaliana*, phylogenetic analysis, *AsaLecRLK* genes were further categorized into three subfamilies, C-type, G-type, and L-type, in *A. sativa*.

### 4.3. Analysis of Motifs, Gene Promotor, Gene Structures, and Conserved Domains

The conserved motifs in the oat LecRLK proteins were determined using the MEME server (https://meme-suite.org/; accessed on 15 February 2024) [27] with a maximum motif number of 10. We used SeqKit v.2.3.0. to extract 1.5 kb sequences upstream of the *LecRLK* genes as promoter regions and submit these sequences to PlantCARE (http://bioinformatics.psb.ugent.be/webtools/PlantCARE/html/; accessed on 23 February 2024) [28] for the analysis of cis-acting regulatory elements. To extract the CDS and UTR locations corresponding to AsaLecRLKs, we used in-house R scripts. The conserved protein domains of AsaLecRLKs were analyzed based on the online software InterPro (https://www.ebi.ac.uk/interpro/entry/pfam/, accessed on 16 February 2024), SMART (https://smart.embl.de/, accessed on 16 February 2024) [65] and NCBI-CDD (https://www.ncbi.nlm.nih.gov/Structure/cdd/ wrpsb.cgi, accessed on 16 February 2024) online databases. And the transmembrane domains were predicted using the Prediction of TMHMM (http://www.cbs.dtu.dk/services/TMHMM-2.0/; accessed on 27 February 2024). Physical mapping of conserved and transmembrane structural domains was performed with in-house R scripts.

### 4.4. Chromosomal Distribution, Gene Duplication, and Ka/Ks Calculation

The chromosomal position information was retrieved from the *Avena sativa* genome sequence files and the corresponding gene structure annotation files. The gene localization of all *AsaLecRLKs* on the chromosome was visualized by TBtools v.2.119 [66].

A gene duplication search for the identified AsaLecRLK members was performed using the MCScanX tool of Tbtools. The rate of synonymous substitutions (Ks) and non-synonymous substitutions (Ka) in the *LecRLK* genes obtained from gene duplication events was calculated using ParaAT 2.0 [67] and KaKs_Calculator v.2.0 [68]. The *AsaLecRLK* gene replications and homologous genetic relationships between species were visualized using the NGenomeSyn v.1.41 [69].

### 4.5. Plant Material, Growth Conditions, and Treatment

Treatment of hexaploid common oat (*Avena sativa*; Accession QL&PHY 498; Riyuehacheng, Huangyuan county, Qinghai Province) seeds involved shaking in 1% NaClO solution for 30 min, followed by rinsing twice with distilled water. The seeds were then germinated on moist filter paper in Petri dishes at 25 °C in darkness. Upon reaching a root length of 3–5 cm (approximately 3–4 days), seedlings were transplanted into polyethylene pots filled with distilled water. Seedlings were secured onto PCR plates floating on distilled water in a growth chamber for two days. The growth chamber conditions were set to 25 °C with a 16 h light/8 h dark cycle and 60% humidity. On the third day, half-strength Hoagland nutrient solution was introduced, with solution renewal every 24 h. When plants reached the two-leaf stage, salt stress was initiated by supplementing the half-strength Hoagland nutrient solution with NaCl to a concentration of 200 mM/L for 0, 6, 12, 24, and 48 h. Fresh root and leaf tissues were immediately frozen in liquid nitrogen and stored at –80 °C for RNA extraction, with three biological replicates per treatment group.

### 4.6. RNA Extraction, cDNA Reverse Transcription, and RT-qPCR Analysis

Leaf and root tissues from *A. sativa* were pulverized into a fine powder using liquid nitrogen for the purpose of RNA extraction, following the procedures outlined in the manual of the FastPure^®^ Cell/Tissue Total RNA Isolation Kit V2 (Vazyme, Nanjing, China). The conversion of RNA into cDNA was facilitated using PrimeScript™ RT Master Mix (Takara, Kusatsu, Japan). Subsequently, the RT-qPCR analysis was conducted using ChamQ Universal SYBR qPCR Master Mix (Vazyme, Nanjing, China). Gene-specific primers, sourced from NCBI, were utilized for this experiment (Appendix A). The ADPR gene served as the normalization control. Gene expression levels were determined through the 2^−∆∆CT^ method, with the control group and the root tissue being normalized to 1 in comparison to the NaCl treatments, as well as among various tissues. Three technical replicates were performed for each biological replicate to ensure the accuracy of the results.

### 4.7. RNA-seq Data Analysis

RNA-seq was performed by the Illumina DNBSEQ platform (carried by BerryGenomics Technology Co., Ltd., Beijing, China). RNA was extracted from the root and leaf of *Avena sativa* using an RNAprep Pure Plant Kit (DP441, TIANGEN, Beijing, China). The library was constructed using the TruSeq RNA v.2 kit (Illumina, San Diego, CA, USA). Transcriptome sequencing was performed by the Illumina platform NovaSeq 6000, and the RNA-seq data can be obtained from the Genome Sequence Archive of the China National Center for Bioinformation, project SRP519844. Low-quality sequences and splice sequences in raw reads were filtered using in-house perl scripts to obtain clean reads. The filtered reads were aligned to the *Avena sativa* cv. ‘Sang’ genome by Hisat2 [70], and the raw number of aligned reads was calculated using the featureCounts v.2.0.2 [71]. The level of gene expression was calculated by determining the number of expected fragments per kilobase of transcript per million mapped reads (FPKM). Heatmaps were generated using the heatmap package in TBtools v.2.119 with log_2_(FPKM + 1) values.

### 4.8. Subcellular Localization

To investigate the functional sites of the *AsaLecRLK* gene family, we randomly selected three genes, *AsaLecRLK-L-type-43*, *AsaLecRLK-L-type-44*, and *AsaLecRLK-G-type-45*, for a subcellular localization experiment. The coding sequence (CDS) without stop codons was cloned into the p1302 vector, which contained a 35S-driven green fluorescent protein (GFP) promoter. The In-Fusion cloning kit named ClonExpress^®^ Ultra One Step Cloning Kit, which is produced by Vazyme, was used for cloning. SpeI was utilized as the restriction site of the p1302 vector. The Agrobacterium containing 35S-AsaLecRLK-L-type-43:GFP, 35S-AsaLecRLK-L-type-44:GFP and 35S-AsaLecRLK-G-type-45:GFP was transiently coinfiltrated into tobacco leaves with a subcellular localization marker. After 48 h of infiltration, the distribution of fluorescence was visualized under a confocal laser scanning microscope (Leica, Biberach, Germany).

## 5. Conclusions

We performed a genome-wide analysis of the *LecRLK* gene family in *Avena* ssp. (*A. sativa*, *A. insularis*, and *A. longiglumis*). In this study, we identified a total of 390 lectin genes in *A. sativa* and analyzed their physicochemical properties, gene structure, conserved domain, phylogenetic relationship, selection pressures, and expression patterns under salt stress. RT-qPCR and subcellular location experiments were used to validate the expression profile of the *AsaLecRLK* genes. *AsaLecRLK-G-type-110* and *AsaLecRLK-L-type-12* genes were considered for further functional characterization. We found that these genes play significant roles in salt tolerance. Our research suggests the involvement of *AsaLecRLK* genes in *A. sativa’s* growth and response to salt stress, providing a theoretical basis for further biological research of oats.

## Figures and Tables

**Figure 1 ijms-25-12754-f001:**
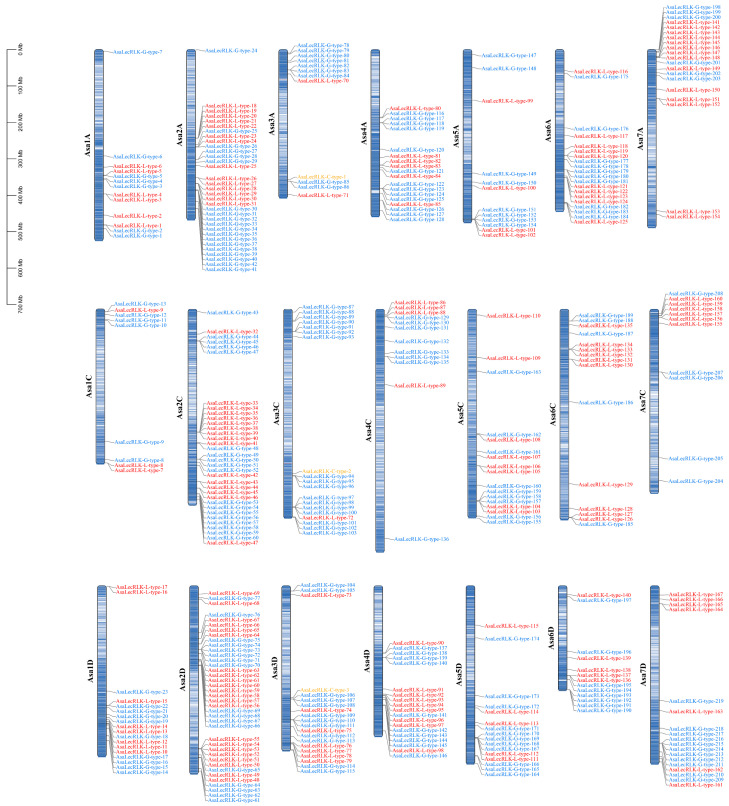
Chromosome location of *LecRLK* gene family of *Avena sativa*; L-type (red), C-type (yellow), and G-type (blue) subfamilies are shown. Chromosome numbers are shown at the left. Center of chromosomes shows the overall gene density. *LecRLK* gene locations are shown on the right.

**Figure 2 ijms-25-12754-f002:**
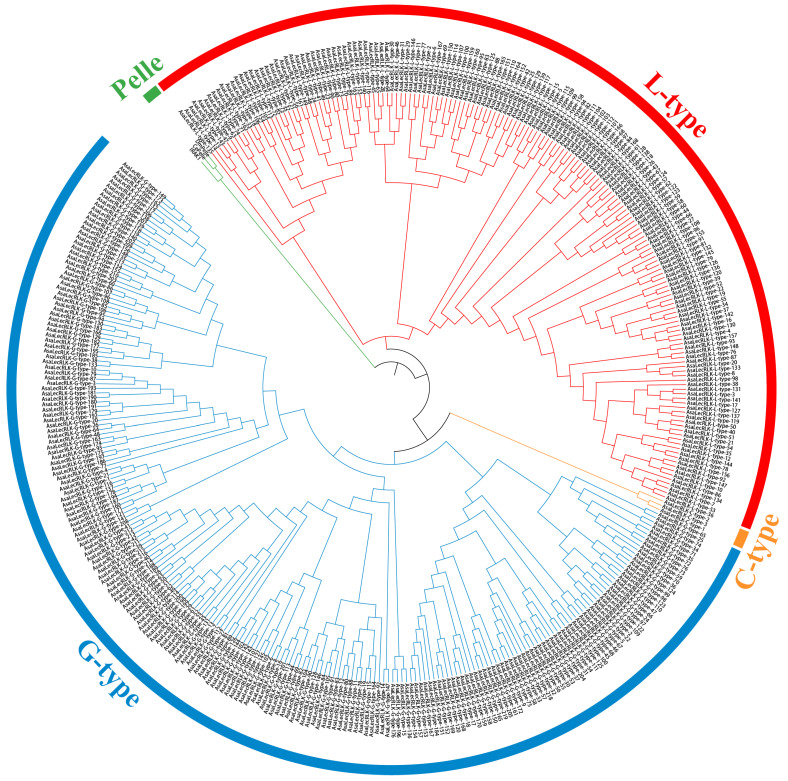
Phylogenetic relationships of LecRLK proteins in *Avena sativa* and three RLK Pelle family proteins in animals. The phylogenetic trees were constructed using the maximum-likelihood method based on predicted protein sequences. L-type (red), C-type (yellow), G-type (blue), Pelle (green).

**Figure 3 ijms-25-12754-f003:**
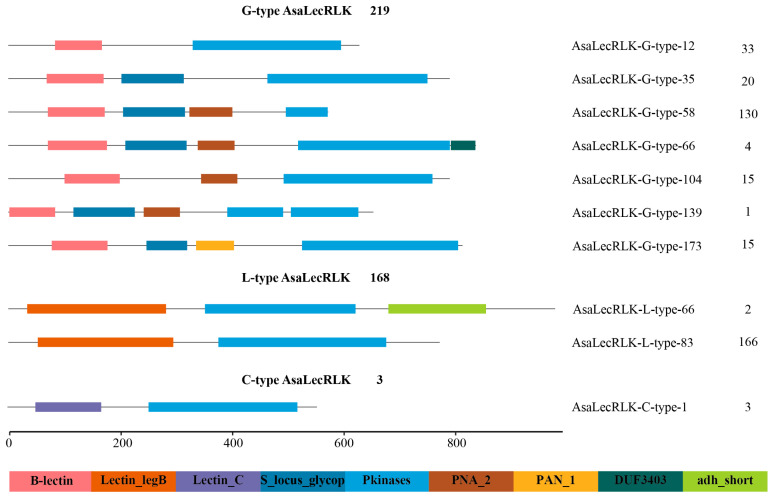
Domain structure prediction of AsaLecRLKs showing the number of genes (right column) with each structure. G-type AsaLecRLKs contain bulb lectin domain, S-locus glycoprotein domain, and PAN domain at the N-terminus and protein kinase domain and DUF3403 domain at the C-terminus; L-type AsaLecRLKs contain the legume lectin domain at the N-terminus and protein kinase domain and adh_short domain at the C-terminus; C-type AsaLecRLK contains the calcium-binding lectin domain at the N-terminus and protein kinase domain at the C-terminus.

**Figure 4 ijms-25-12754-f004:**
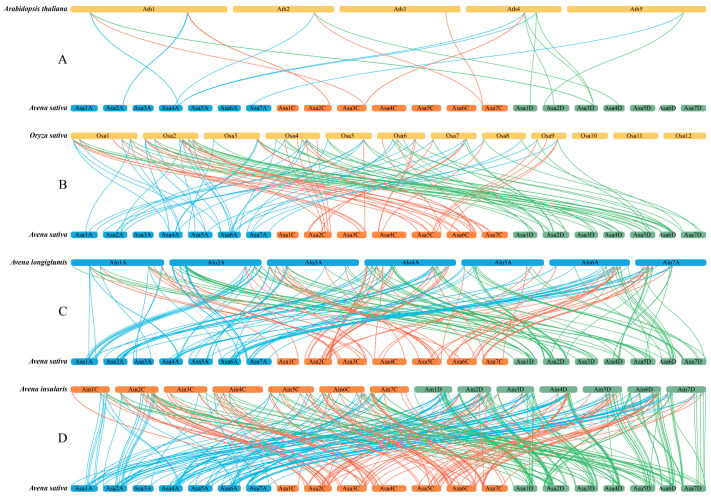
Synteny analyses of *LecRLK* gene family between *Arabidopsis thaliana* and *Avena sativa* (**A**), *Oryza sativa* and *A. sativa* (**B**), *A. longiglumis* and *A. sativa* (**C**), and *A. insularis* and *A. sativa* (**D**). Lines represent collinear gene pairs between genomes of *A. sativa* and other species. Blue line: A genome; red line: C genome; green line: D genome.

**Figure 5 ijms-25-12754-f005:**
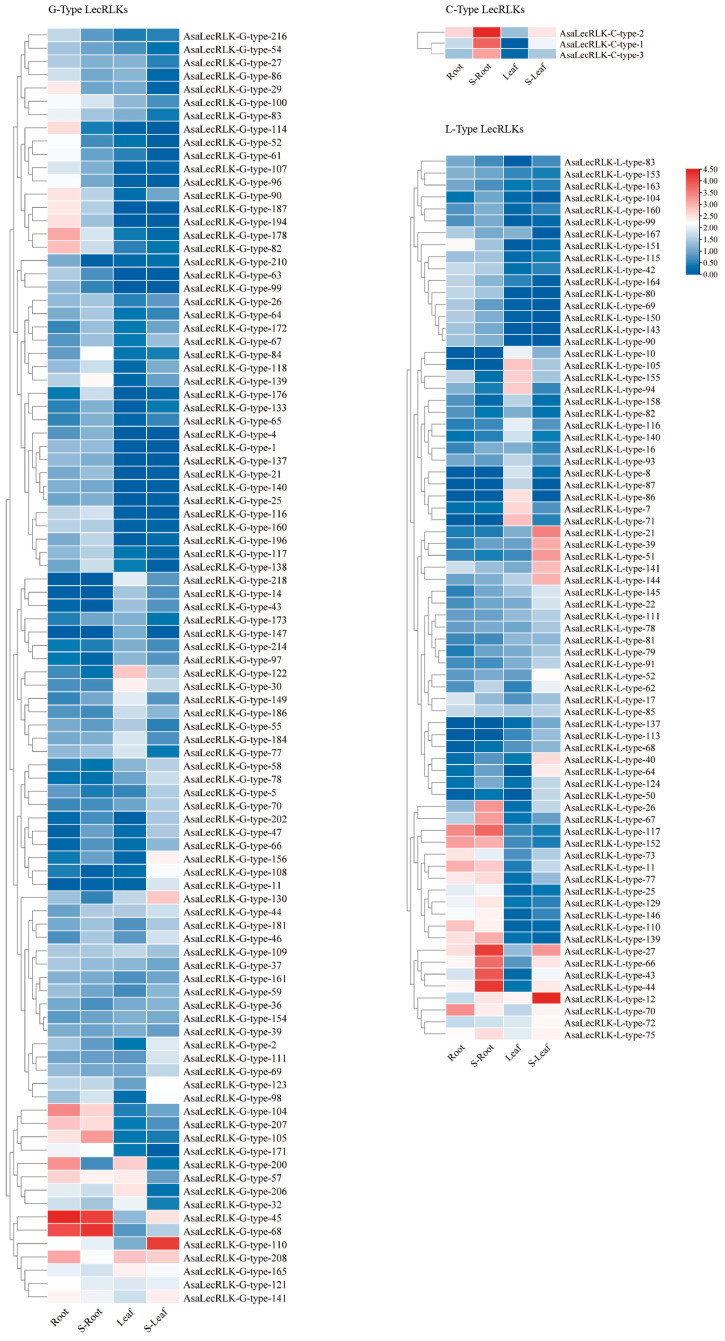
Expression profiles of *AsaLecRLK* genes under different levels of salt stress in root and leaf. RNA-sequencing data on salt stress for *AsaLecRLKs*. The heatmap was generated on the Log2 of (FPKM+1) values using TBtools. Color bar represents normalized FPKM values: red, high expression level; blue, low expression level.

**Figure 6 ijms-25-12754-f006:**
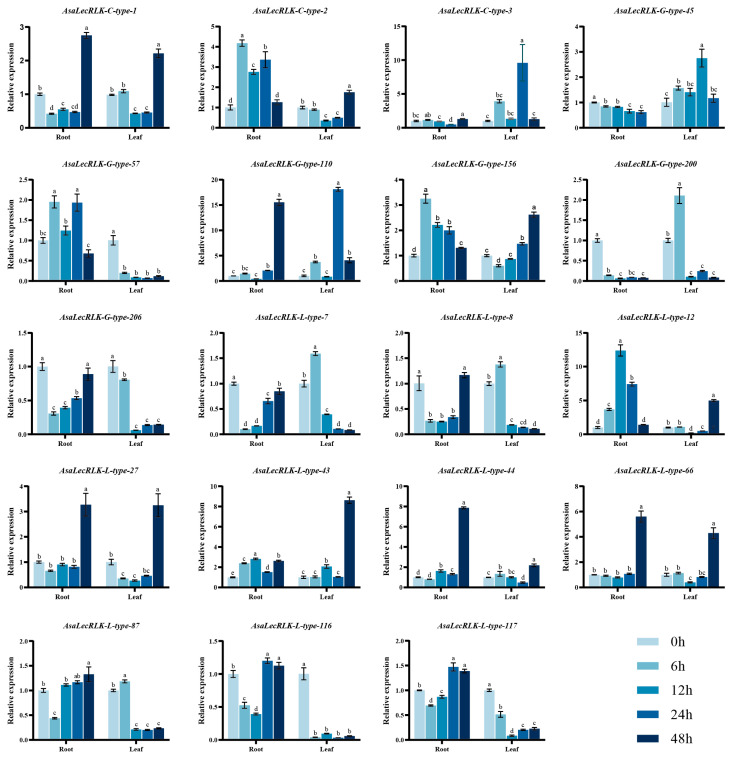
Relative expression level of *AsaLecRLK* genes under salt stress after 0 h, 6 h, 12 h, 24 h and 48 h in root and leaf tissue. Expression level of each gene at 0 h is set as reference. The data represent the mean values of three replicates ± SD. Statistical significance of differences was tested by one-way ANOVA analysis (*p* < 0.05) and is indicated by lowercase letters.

**Figure 7 ijms-25-12754-f007:**
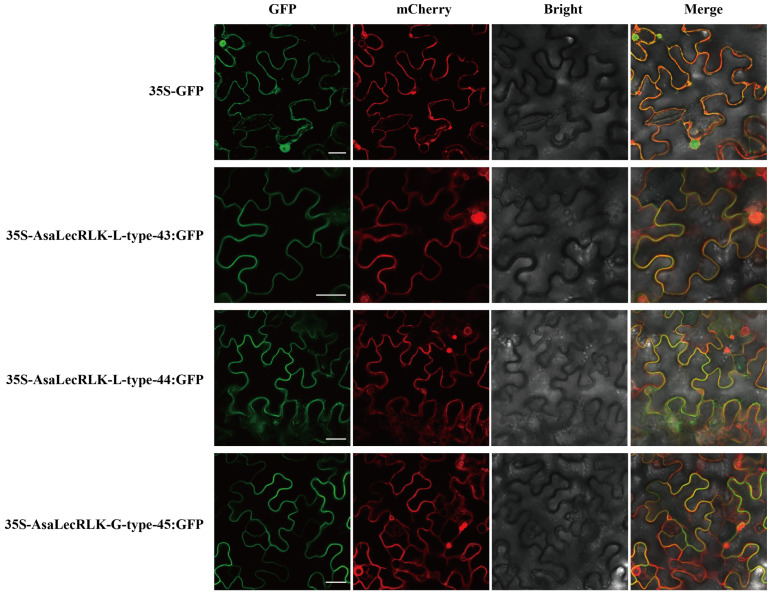
The subcellular localization of *AsaLecRLK-L-type-43*, *AsaLecRLK-L-type-44*, and *AsaLecRLK-G-type-45*. Bars = 25 μm. The figures show confocal images of GFP fluorescence, plasmalemma localization (mCherry), bright field, and composite field.

## Data Availability

Data are contained within the article and Appendix A.

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
