# Peer review of "Genome-Wide Identification of the Lectin Receptor-like Kinase Gene Family in Avena sativa and Its Role in Salt Stress Tolerance"

_ijms, 2024, doi:10.3390/ijms252312754_

Round 1

Reviewer 1 Report

Comments and Suggestions for Authors

The authors thoroughly investigate changes in metabolism as plants respond to salt stress. These changes are important because they could be correlated with drought stress, i.e., losing water from the cell. The research topic is very interesting and could benefit to the scientific community in more productive harvest and tastier crops. The Lectin receptor-like kinase plays an important role in oat species metabolism, hence understanding in changes of that enzyme can lead to a better understanding the whole metabolism. The introduction is concise and precise and the authors describe receptor-like kinase and all the potential binding sites to which LecRLKs could interact. In addition, they also pointed out various interactions as responses to environmental conditions. The article is well organized with clear focus on the research framework analyzing all-important points that could affect experiments and results.

In my opinion additional improvements could be regarding the contribution of the other enzymes  that participate in the metabolism and could alter the plants' response.

materials and methods are explained very well and the experimental conditions are described precisely.

the presentation of the results follows the logical pattern and can be easily analyzed. the discussion of thoroughly analyzed presented results and advantages and disadvantages of the method is pointed out.

The conclusion summarised presented and discussed results underlining the most important.

The references are appropriate for the article but a little bit difficult to follow hence I sugest authors to separate them from the rest of the text with brackets.

in my opinion the number  of figures is too big especially figures with similar content and look, which could confuse the reader and lead them in wrong direction. hence I suggest authors to show just important part or part where the difference is noticed . at current stage the  figures 1,2,4,5,6 are little bit fuzzu and confusing.

Also reducing the number of figures i.e.  keep just the most important and viewable figures could also bring benefit to the reading and understanding the article.

Author Response

Reviewer 1

The research topic is very interesting and could benefit to the scientific community in more productive harvest and tastier crops. The lectin receptor-like kinase plays an important role in oat species metabolism, hence understanding in changes of that enzyme can lead to a better understanding the whole metabolism. The introduction is concise and precise and the authors describe receptor-like kinase and all the potential binding sites to which LecRLKs could interact. In addition, they also pointed out various interactions as responses to environmental conditions. The article is well organized with clear focus on the research framework analyzing all important points that could affect experiments and results.

Reply: Reviewer 1 has been appreciated for the positive comments on the manuscript. The careful modifications have been provided in order to meet reviewer’s suggestions and the journal requirements.

  1. In my opinion additional improvements could be regarding the contribution of the other enzymes that participate in the metabolism and could alter the plants’ response.

Reply: Thanks for the reviewer 1’s insightful comments. We have carefully discussed the potential contribution of other enzymes involved in metabolism that could influence the plants’ response to salt stress, please see lines 371–383 in the discussion section of the revised manuscript.

  1. Materials and Methods are explained very well and the experimental conditions are described precisely.

Reply: Thanks for your positive comments on the Materials and Methods section. Your opinion is critically important for the objective evaluation of our paper.

  1. The presentation of the results follows the logical pattern and can be easily analyzed. The discussion of thoroughly analyzed presented results and advantages and disadvantages of the method is pointed out.

Reply: Thanks for your positive comments on the result presentation and method evaluation. The comments are critically important for the objective evaluation of our paper.

  1. The conclusion summarized presented and discussed results underlining the most important.

Reply: Thanks for your positive comment on the conclusion section.

  1. The references are appropriate for the article but a little bit difficult to follow hence I suggest authors to separate them from the rest of the text with brackets.

Reply: Thanks for your suggestion. We thoroughly checked the whole manuscript and added one space between the citation number with brackets from the text, for example, Line 395, “homeostasis and normal cell function [50]”, in the revised manuscript.

  1. In my opinion the number of figures is too big especially figures with similar content and look, which could confuse the reader and lead them in wrong direction. Hence, I suggest authors to show just important part or part where the difference is noticed. At current stage, the figures 1,2,4,5,6 are a little bit of fuzzy and confusing.

Reply: Agree. Figure 4 has been moved to the supplementary Figure S4 in order to keep the main text figures being more concise and focused. Additionally, we have carefully updated the figure and supplementary figures’ number and he corrected presence in the revised manuscript.

  1. Also reducing the number of figures, i.e., keep just the most important and viewable figures could also bring benefit to the reading and understanding of the article.

Reply: Thanks for your consideration. Figure 4 has been moved to the supplementary Figure S4 in order to keep the main text figures being more concise and focused. Additionally, we have carefully updated the figure and supplementary figures’ number and he corrected presence in the revised manuscript.

Reviewer 2 Report

Comments and Suggestions for Authors

This paper presents an interesting, complex study with practical applications about the identification and characterization of the lectin receptor-like kinases (LecRLKs) gene family in Avena sativa and their roles in salt stress tolerance. The authors identified LecRLKs Gene Family Members in Oat, showed the Chromosomal Distribution and phylogenetic analysis of these genes, analyzed the gene structure of the AsaLecRLKs gene family and demonstrated the AsaLecRLKs response to salt stress treatments. The materials and methods used are presented clearly and concisely.

Observations

- In the Discussion section, it would be necessary to show the results of similar studies for other crop plants regarding salt stress tolerance;

- Line 458 – "A. thaliana" in italics

- It would have been interesting if the plants' resistance to salt stress had been checked in soils with the same concentration of salt.

- The writing of bibliographic references in the text should be in accordance with the writing instructions.

Author Response

Reviewer 2

This paper presents an interesting, complex study with practical applications about the identification and characterization of the lectin receptor-like kinases (LecRLKs) gene family in Avena sativa and their roles in salt stress tolerance. The authors identified the LecRLKs gene family members in oat, showed the chromosomal distribution and phylogenetic analysis of these genes, analyzed the gene structure of the AsaLecRLKs gene family and demonstrated the AsaLecRLKs response to salt stress treatments. The materials and methods used are presented clearly and concisely.

Reply: Thanks for your positive comments on the result presentation and material and method evaluation. The comments are critically important for the objective evaluation of our paper.

  1. In the discussion section, it would be necessary to show the results of similar studies for other crop plants regarding salt stress tolerance.

Reply: Agree, the comparisons with the similar studies on salt stress tolerance of other crop species (like rice and soybean) have been provided, please see Lines 384–397 in the discussion section of the revised manuscript.

  1. Line 458—"A. thaliana” in italics.

Reply: Thanks so much for your careful suggestion. We have corrected A. thaliana to be italic in Line 474 of the revised manuscript.

  1. It would have been interesting if the plants’ resistance to salt stress had been checked in soils with the same concentration of salt.

Reply: Thanks so much for your insightful suggestion. In our preliminary experiment, the transcriptome data was obtained from the soil cultivation seedling study. However, due to the accuracy limit of soil retention salinity and the potential influence of soil micro-organisms on gene expression under salt stress, consequently, we have to adopt hydroponic conditions for the seedling study at the second experiment. Hydroponics allowed for the more precise salinity control of experimental conditions and ensured higher reproducibility of the transcriptome data. Additionally, the seedling studies are consistently for materials of transcriptome analysis and those of RT-qPCR experimental validation.

  1. The writing of bibliographic references in the text should be in accordance with the writing instructions.

Reply: Agree, thanks for the suggestion. We have carefully reviewed all bibliographic references in order to ensure each one of them following to the journal’s instructions.

Reviewer 3 Report

Comments and Suggestions for Authors

Paper deals with the genome -wide analysis of LecRLKs gene family in Avena ssp. The results are valuable in understanding the growth and response of A.sativa to salt stress. The paper is quite well written with a high scientific level. Methods were properly chosen whilst results are described quite clearly and are followed by the discussion. However I found a few weaknesses that should be improved before publication. My detailed comments are listed below:

1 – Numbers of citing positions should be separated from the text in some way (e.g., using brackets) because they make the text more difficult to understand in a present form.

2 – Figure 3 – authors should refer to this figure in the text of the manuscript

3 – Figure S2 and S3 – I didn’t find them in supplementary materials. It should be improved.

Author Response

Reviewer 3

Paper deals with the genome-wide analysis of LecRLKs gene family in Avena ssp. The results are valuable in understanding the growth and response of A. sativa to salt stress. The paper is quite well written with a high scientific level. Methods were properly chosen whilst results are described quite clearly and are followed by the discussion. However, I found a few weaknesses that should be improved before publication.

Reply: Thanks so much for your insightful suggestions. We have addressed the weaknesses you raised carefully and provided the point-by-point response to your suggestions in the revised manuscript.

  1. Numbers of citing positions should be separated from the text in some way (e.g., using brackets) because they make the text more difficult to understand in a present form.

Reply: Thanks so much for your suggestion. We have added the space before the citation numbers with brackets in order to make the text understandable in the revised manuscript.

  1. Figure 3, authors should refer to this figure in the text of the manuscript.

Reply: Thank you for the suggestion. We have already referred to Figure 3 in Lines 143 and 351 of the revised manuscript twice, i.e., “Figures 3, S1”.

  1. Figures S2 and S3—I didn’t find them in supplementary materials. It should be improved.

Reply: Thanks so much for your enquiry. Both Figures S2 and S3 have been provided in the supplementary materials. Additionally, Figure S2 has been cited in Line 167, Figure S3 has been cited in Lines 177, 356 and 406 of the revised manuscript.